# Experimental Investigation on the Effect of Graphene Oxide Additive on the Steady-State and Dynamic Shear Properties of PDMS-Based Magnetorheological Elastomer

**DOI:** 10.3390/polym13111777

**Published:** 2021-05-28

**Authors:** Minzi Liu, Mei Zhang, Jiangtao Zhang, Yanliang Qiao, Pengcheng Zhai

**Affiliations:** Hubei Key Laboratory of Theory and Application of Advanced Materials Mechanics, School of Science, Wuhan University of Technology, Wuhan 430070, China; liuminzi@whut.edu.cn (M.L.); QiaoYanliang@whut.edu.cn (Y.Q.); pczhai@126.com (P.Z.)

**Keywords:** magnetorheological elastomers, graphene oxide, magnetorheological effect, steady-state shear, dynamic shear property

## Abstract

Isotropic polydimethylsiloxane (PDMS)-based magnetorheological elastomers (MREs) filled with various contents of graphene oxide (GO) additive were fabricated by the solution blending-casting method in this work. The morphologies of the produced MREs were characterized, and the results indicate that the uniform distribution of GO sheets and carbonyl iron particles (CIPs) becomes difficult with the increase of GO content. The steady-state and dynamic shear properties of the MREs under different magnetic field strengths were evaluated using parallel plate rheometer. It was found that the physical stiffness effect of GO sheets leads to the increase of the zero-field shear modulus with increasing GO content under both the steady-state and dynamic shear loads. The chemical crosslinking density of PDMS matrix decreases with the GO content due to the strong physical crosslinking between GO and the PDMS matrix. Thus, the MREs filled with higher GO content exhibit more fluid-like behavior. Under the dynamic shear load, the absolute MR effect increases with the GO content due to the increased flexibility of the PDMS matrix and the dynamic self-stiffening effect occurring in the physical crosslinking interfaces around GO sheets. The highest relative MR effect was achieved by the MREs filled with 0.1 wt.% GO sheets. Then, the relative MR effect decreases with the further increase of GO content due to the improved zero-field modulus and the increased agglomerations of GO and CIPs. This study shows that the addition of GO sheets is a possible way to prepare new MREs with high MR effect, while simultaneously possessing high zero-field stiffness and load bearing capability.

## 1. Introduction

MREs are typically produced by adding micro-sized soft magnetic particles into a polymeric matrix followed by mechanical blending and curing. Since the magnetic particles are locked in the polymeric matrix, the sedimentation and leakage problems associated with their liquid-state analog, i.e., magnetorheological fluid (MRF), are overcome. When the cured MREs are exposed to an external magnetic field, the magnetic particles tend to be compactly packed and form chain-like network structures parallel to the magnetic field as a result of the magnetic dipolar attractions between the magnetized particles [1,2]. The magnetized particle network can also serve as a reinforcing frame and offers restriction to the mobility of the polymer chains [2,3,4], thus altering the stiffness and damping properties of MREs. Compared with MRFs, MREs have more rapid response to the magnetic field, better structural stability, simpler design and lower preparation cost. Furthermore, the mechanical performances of MREs can be continually and reversibly adjusted by an applied magnetic field. These advantages make them highly promising for many future intelligent systems, such as smart damper [5,6], sensor [6,7,8], actuator [9,10], vibration isolators [11,12] and multi-functional electronic skin [13].

The change in the mechanical performance of MREs with the external magnetic field is referred to as the magnetorheological (MR) effect. The relative MR effect is defined by the ratio of the magnetic field induced change in the mechanical performance to the zero-field performance. High relative MR effect is always desired in the design of MRE-based intelligent systems, since it indicates a larger range of controllability [6,14,15]. In principle, the high MR effect of MREs can be achieved by using soft matrices [15,16,17]. However, the produced MREs with soft matrices have low zero-field moduli, which is in most cases undesirable from a practical point of view [15,18,19]. The higher particle fraction can also lead to a larger increment in the compressive modulus [20], dynamic storage and loss moduli [21]. A high particle fraction in MREs causes a great increase of the viscosity of the pre-cured composite and particle aggregations, thus doing harm to the properties of MREs [2,22]. Thus, it currently remains a great challenge to develop new MREs with high MR effect, while maintaining proper zero-field stiffness [16].

It has been found that the additives can average the distribution of internal stress in the materials, making them ideal for modifying MREs [8,23]. Conventional additives including carbon black (CB) [11,18,24,25], carbon nanotube (CNT) [15,17,26,27,28] and graphite [8,29] have been used in the fabrication of MREs. Khimi et al. [11] prepared 30, 50 and 70 phr CB-filled anisotropic MREs composed of silane-modified iron particles and a natural rubber (NR) matrix. They found that the addition of CB constrained the movement of iron particles, the chain-like columnar structures became shorter and CB-filled MREs showed the higher energy absorption than that without CB. Chen et al. [24] prepared CBN330-filled MREs, showing that the addition of CB leads to a well-bound microstructure and results in high MR effect, low damping ratio and improved tensile strength. Tian et al. [8,29] found that the addition of graphite particles in PDMS-based MREs results in an increment of the absolute MR effect but a decrement of the relative MR effect due to the increased zero-field mechanical properties. Aziz et al. [17] investigated the magnetic field dependent rheological properties of the NR-based MREs filled with functionalized multiwalled CNT (COOH-MWCNT). The results reveal that the addition of COOH-MWCNT helps to improve the initial modulus, magneto-induced modulus and damping of MREs, and these viscoelastic properties begin to decrease at a COOH-MWCNT content higher than 1.0 wt.%. Li et al. [15,26] and Poojary et al. [27] investigated the effect of CNT on the dynamic mechanical responses of silicone rubber (SR)-based MREs and found that the nanocomposites showed higher zero-field stiffness and damping but a lower relative MR effect compared to the MREs without CNT.

Over the past few years, much attention has been paid to the polymer nanocomposites filled with graphene and its derivatives because of their good conductivities of heat and electricity, large elastic modulus and high specific surface area [1,30,31,32]. Thus, graphene and its derivatives represent rational alternatives to conventional additives. Li et al. [1] found that, by modifying the MREs with graphene nanoplatelets (GNPs), the adhesion force of PDMS-based MREs was greatly improved in both the absence and presence of an external magnetic field. Bica et al. [31,32] investigated the SR-based MREs filled with GNPs to enhance their electrical conductivity and electrical capacitance. However, maximizing the reinforcing efficiency of GNPs in elastomers is still confronted with two challenges: poor dispersion of GNPs in the elastomers and weak interfacial interaction between GNPs and elastomers [33,34].

As one of the most important derivatives of graphene, GO has abundant functional groups, such as hydroxyl, carboxyl and epoxy, which make GO uniquely amphiphilic and result in the uniform distribution of GO sheets in different polymeric matrices. These functional groups also facilitate the good interaction between the polymer matrix and GO sheets, leading to the physical reinforcement and the improvement of the mechanical, thermal and electromagnetic properties of polymer matrix nanocomposites [35,36,37,38,39,40,41]. On the other hand, GO is a two-dimensional material. It is different from fibril-like CNTs and point-like particles, which are filled in the gaps between the matrix molecules. GO sheets can separate the reacting moieties of polymer chains from each other and inhibit their chemical crosslinking reactions [31,42]. This indicates that the addition of GO into MREs can improve the flexibility of the polymer matrix around the magnetic particles, thus endowing the magnetic particles with great mobility under the magnetic field and improving the MR effect of the MREs. In summary, the addition of GO sheets in MREs may be a possible way to prepare new MREs with high MR effect, while simultaneously maintaining high zero-field stiffness. However, studies quantifying the impact of GO sheets on the MR effect of MREs are rarely reported. A deep understanding on the correlation between GO and the MR effect of MREs is important to evaluate the use of GO as a new additive, which was the focus of this study.

We conducted experimental studies to discover the effect of the GO additive on the steady-state and dynamic shear properties of PDMS-based MREs. Series of isotropic PDMS-based MREs filled with different contents of GO sheets were fabricated by a solution blending-casting method. Then, the microstructures of the GO sheets and the produced MREs were analyzed by atomic force microscopy (AFM) and scanning electron microscopy (SEM), respectively. The steady-state shear stress–strain curves and the dynamic rheological behavior of the GO-filled MREs under different magnetic field strengths were tested by using a parallel plate rheometer, and the influence of GO content and the plausible mechanisms were considered.

## 2. Material and Fabrication of MREs

### 2.1. Materials

A commercially available PDMS (Sylgard 184 Silicone Elastomer, Dow Corning Corp., Midland, MI, USA) was used as the matrix and supplied as two-part liquid elastomer kit. Part A contains the pre-polymer base and Part B provides the crosslinking agent. The recommended mass ratio of Parts A and B is 10:1. The GO powder was purchased from Chengyi Education Science & Technology Co. Ltd., Wuxi, Jiangsu, China, with the diameter of 0.2–20 μm. CIPs with an average diameter of 2–3 μm were purchased from Xuliheng New Material Co. Ltd., Xianyang, Shanxi, China. Tetrahydrofuran (THF) was produced by Sinopharm Chemical Reagent Co. Ltd., Beijing, China. All of these materials were used as received.

### 2.2. Fabrication of Isotropic GO-Filled MRE Samples

The fabrication procedure of the isotropic GO-filled MREs is presented schematically in Figure 1. Firstly, a predetermined amount of GO was dispersed into THF solution in a beaker by ultrasonication in a water bath combined with vigorous stirring using a high shear emulsifying mixer for 1 h. Then, the PDMS pre-polymer (Part A) was added into the GO/THF suspension. The mixture was subjected to magnetic stirring at room temperature for 1 h to adequately dissolve the pre-polymer base in the GO/THF suspension. Subsequently, the temperature rose to 70 °C while keeping magnetic stirring. The mixture was weighed every half an hour until its mass no longer changed. It took 4 h to evaporate the THF solvent in the resulting mixture. After cooling to room temperature, the CIPs were added into the emulsion of GO/PDMS pre-polymer, followed by stirring vigorously using a mechanical stirring machine for 2 h to adequately disperse the CIPs. Then, the crosslinking agent (Part B) was added into the mixture, followed by mechanical stirring for another 10 min. The final mixture was poured into the copper molds and degassed in a vacuum box for 1 h. The GO-filled MRE samples were finally obtained after curing in an oven at 60 °C for 12 h.

Four isotropic PDMS-based MREs filled with the GO fractions of 0, 0.1, 0.3 and 0.5 wt.% were produced, which are referred to here as MRE-0, MRE-1, MRE-3 and MRE-5, respectively. The mass fraction of CIPs in the MREs was 60%. Here, the highest GO fraction was 0.5 wt.% because more GO sheets made the GO/PDMS pre-polymer emulsion highly viscous, thus preventing CIPs from uniformly dispersing in the emulsion and the GO-filled MREs from solidifying.

### 2.3. Microstructure Characterization

AFM (NanoMan vs. from Bruker Corporation, Berne, Switzerland) was used to characterize the morphology of GO sheets. GO/THF suspension was deposited on a freshly cleaved mica wafer by spin coating and dried at room temperature. AFM images were obtained using a tapping mode under ambient condition, and the height profile of the GO sheets was measured. Three replicas of AFM were tested. SEM (SU8010, Hitachi Corporation, Osaka, Japan) was utilized to characterize the microstructures of the GO-filled MREs with an acceleration voltage of 20 kV. The samples were cryo-fractured in liquid nitrogen and the fractured surfaces were sputtered by a thin layer of gold powders for SEM observation.

### 2.4. Mechanical Property Tests

To estimate the effect of GO content on the steady-state and dynamic MR effects of the produced MREs, the steady-state shear stress–strain curves, dynamic shear storage modulus (G’) and loss factor (tan(δ)) were measured by using the rotational rheometer equipped with an electromagnetic kit. The electromagnetic kit can generate a magnetic field perpendicular to the direction of the shear flow. A magnetic flux density ranging from 0 to 900 mT can be generated by varying the current in the range from 0 to 3 A. A 25 mm diameter parallel-plate measuring system with 1 mm gap was used. The samples were clamped between the plates with a normal force of 10 N applied to the samples to guarantee a contact between the sample and plates. Each sample was tested in the steady-state rotary shear and oscillatory shear modes to investigate its steady-state and dynamic MR properties, respectively.

For the steady-state shear tests, the shear stress–strain curves were measured at the shear strain rate of 0.005 s^−1^ up to the total strain of 60%. The process of the steady-state shear loading was repeated for each sample at a zero-field condition until a convergent stress–strain curve was obtained. Thus, the stress softening (the Mullins effect) was excluded from the stress–strain curves. Before each repeated loading process, the shear load generated in the previous loading process was released to zero and the sample was allowed to stand for 2 min. Then, the desired magnetic field was applied and the stress–strain curves were tested. For the dynamic magnetorheological tests, the samples were also preconditioned at a zero-field condition using the same cyclic loading process as that for the steady-state tests. The strain amplitude sweeps were carried out at the constant frequency of 1 Hz with the strain amplitude varying from 0.01% to 100%. The frequency sweeps were carried out at the constant strain amplitude of 0.1% with the frequency varying from 0.1 to 100 Hz. For both the steady-state and dynamic tests, the applied magnetic flux density increased from 0 to 900 mT in 150 mT increments. All results were averaged based on three tests for each condition.

## 3. Results and Analyses

### 3.1. Microstructure

The typical AFM image and the corresponding height profile of GO sheet are shown in Figure 2. It can be seen that the GO sheets have the average thickness about 4.33 nm, indicating the graphene sheets have 3–4 layers.

The SEM images of the produced MREs are shown in Figure 3. It can be seen that MRE-0 has a uniform distribution of CIPs without obvious aggregation (Figure 3a). GO aggregations can be occasionally observed in MRE-1 and MRE-3 (Figure 3b,c). There are also some GO sheets with the size over 10 μm and the micro-fissures caused by pulling out of GO sheets present in the fracture surfaces, as illustrated in Figure 3b,c. However, GO sheets with smaller size can rarely be observed in the fractured surfaces, indicating they are deeply embedded in the matrix. In the fracture surface of MRE-5, however, the GO aggregations and micro-fissures caused by pulling out of GO sheets become more serious. These results indicate that the restack of the exfoliated GO sheets happens in the GO/PDMS pre-polymer solution during the stirring and THF evaporation processes, and it increases with the increase of GO content due to the high hydrophilicity of GO [42,43,44]. Figure 3e shows a magnified view of the fractured surface of MRE-5. It can be observed that some small CIPs are absorbed on the GO surfaces or aggregated near the GO sheets, which may be caused by the strong adsorption between GO sheets and CIPs. Zhang et al. [41] also found that the micron-sized GO plates can adhere on the surface of CIPs and fulfill the free interspace among CIPs in the CIPs/GO mixture-based MRFs, thus enhancing the viscosity and improving the sedimentation problem of MRFs. The increased viscosity of the GO/PDMS emulsion with the increase of GO content (see Appendix A) was also found in this work. However, this makes uniformly distributing CIPs during the fabrication process difficult.

### 3.2. Steady-State Shear Tests

#### 3.2.1. Result Analysis

Figure 4 demonstrates the typical steady-state shear stress–strain curves of MRE-0 under four consecutive loading cycles in the zero-field condition. It can be seen that a stress softening phenomenon (i.e., the Mullins effect) occurs from the first to the third loading process, and a convergent stress–strain curve is obtained during the fourth loading process. It is widely recognized that the Mullins effect occuring in particle-filled polymer composites under cyclic loading is attributed to several factors, such as the chain slipping and disentanglement of polymer molecules, slippage and separation of polymer chains from the particle surfaces and the breakdown of the particle aggregates [45,46]. The stress overshoot in the stress–strain curve of the first loading process is caused by the static friction and sliding between sample and upper plate surface [2,23]. The convergent stress–strain curve’s slope decreases gradually with the increase of shear strain, and the stress finally reaches a plateau. This is caused by the continuous slippage and separation of PDMS chains from the particle surfaces and the slipping and breaking of PDMS chains [23,45,47].

Figure 5 shows the steady-state shear stress–strain curves of the produced MREs under different magnetic fields. The tests revealed that all samples are magnetically saturated at the magnetic flux density above 750 mT, as shown in Figure 5a. Thus, the test results at 900 mT are not given in Figure 5c,d. As shown in Figure 5, all curves show stress plateaus under different magnetic fields.

To characterize the effect of GO content on the mechanical properties of the MRE samples, shear moduli (G_T_) at different magnetic fields were calculated based on the linear slope between 1% and 2% shear strain. An approximation of the first derivative of the stress–strain curves is acceptable using this small strain increment in this small strain range [48]. The values of shear moduli G_T_ and plateau stress σ_p_ of the MREs at different magnetic fields are compared in Figure 6. Figure 6 shows that all MREs present enhancements in both G_T_ and σ_p_ with increasing the magnetic field strength, indicating that all MRE samples exhibit obvious MR effects. The MR effect is usually explained by the increased particle magnetization and the enhanced inter-particle magnetic dipolar attractions with increasing the magnetic field [2,3,4]. Furthermore, the increased inter-particle magnetic dipolar attractions can lead to the improved movement of the particles to align with the magnetic field direction, thus inducing the strong reinforcement effect of the particles [1,2,3,6,17,27,45].

Figure 6a shows that in the zero-field state, the G_T_ of MRE-0 is 113.5 kPa and the GO-induced increments in G_T_ are 35.3, 94.4 and 171.2 kPa for the samples filled with 0.1, 0.3 and 0.5 wt.% GO, respectively. These increments only result from the physical strengthening effect of GO sheets, and they increase with GO content. In the magnetic saturation state, the G_T_ of MRE-0 is increased to 171.9 kPa due to the MR effect. The GO-induced increments in G_T_ are 54.6, 89.1 and 176.0 kPa for the samples with 0.1, 0.3 and 0.5 wt.% GO, respectively. Comparing with those in the zero-field state, only MRE-1 shows the obvious improvement in the GO-induced increments in G_T_ in the magnetic saturation state. Since the increment in G_T_ in the magnetic saturation state is due to both the physical strengthening effect and the modified MR effect resulting from the loading of GO sheets, the result indicates that only MRE-1 shows an improved MR effect compared with MRE-0. The further increase of GO content causes a decrease in the MR effect.

To quantitatively analyze the effect of GO content on the MR effect of the MREs under the steady-state and dynamic shear loads, both the absolute MR effect ΔP and relative MR effect $P_{r}$ were calculated by [2,6,17]:(1)$$\Delta P=P_{sat}-P_{0}$$
(2)$$P_{r}=\Delta P/P_{0}\times 100\%$$
where *P* represents the steady-state shear modulus G_T_, the plateau stress σ_p_ or the dynamic storage modulus G’. The subscripts “sat” and “0” represent the magnetic saturation and zero magnetic field, respectively.

The absolute and relative MR effects of the MREs under steady-state shear load are shown in Figure 7. It is obvious that MRE-1 has the highest MR effect among the produced MREs for both G_T_ and σ_p_. The absolute MR effect of G_T_ is improved by 19.3 KPa (33.0%) from 58.4 KPa for MRE-0 to 77.7 KPa for MRE-1, while the relative MR effect is improved by only 0.8% from 51.4% to 52.2% due to the improved zero-field modulus of MRE-1. The further increase of GO content results in a decrease of both the absolute and relative MR effects due to the increased aggregations of GO and CIPs. As for the effect of GO content on σ_p_, it shows the same tendency as G_T_, but the increment in σ_p_ resulting from the GO filling is much less than that in G_T_. The absolute MR effect of σ_p_ is improved by 5.39 KPa (117.4%) from 4.59 KPa for MRE-0 to 9.98 KPa for MRE-1 and the relative MR effect is improved by 18.7% from 19.5% to 38.2%.

#### 3.2.2. Microstructural Analysis on the Effect of GO Sheets

The different effects of GO filling on G_T_ and σ_p_ are caused by the interfacial interaction between the GO sheets and PDMS matrix and the microstructural deformation mechanisms, as illustrated in Figure 8. It has been reported that strong physical crosslinking interfaces exist between the GO sheets and PDMS matrix due to the strong van der Waals forces, hydrogen bonding and CH–π interactions [1,30,43,44], which is also proved indirectly by the viscosity tests on the GO/PDMS pre-polymer emulsion, as given in Appendix A. The strong interfacial physical interactions can induce a three-dimensional GO network over the MRE samples, as illustrated in Figure 8a. Under the small strain state, the physical crosslinking interfaces are intact (Figure 8b). Thus, the GO sheets impede the slipping between PDMS chains and lead to the strong physical strengthening effect [6,16,44]. On the other hand, this physical strengthening effect causes the inner stress in the physical crosslinking interfaces around GO sheets to accumulate rapidly. Nevertheless, the physical crosslinking interfaces between the PDMS matrix and GO sheets are much weaker than those of the chemical cross-linked PDMS chains, thus they are more likely to slip and break under stress. This in turn results in an enhanced stress softening with the increase of GO content under the large deformation. As shown in Figure 5, the strain at which the significant stress softening happens decreases obviously with the increase of GO content, as does the strain at which the stress reaches a plateau.

In the stress plateau region, most of the physical crosslinking interfaces between the GO sheets and PDMS matrix were broken, and the interface voids developed with the increase of the shear deformation, as illustrated in Figure 8c. Then, the physical strengthening effect of GO sheets is greatly reduced, and the main load-carrying component is the PDMS matrix in the form of the chain tension and friction. In the GO-filled MREs, the chemical crosslinking of PDMS chains is inhibited due to the separated reacting moieties of PDMS chains from each other by GO sheets and the strong physical crosslinking between the GO sheets and PDMS chains [30,44]. The low chemical crosslinking density results in an improved flexibility and a reduced modulus of the PDMS matrix [2,49]. Moreover, the increase of GO content leads to the agglomerations of GO and CIPs, as revealed in Section 3.1, which also cause the reduction in the reinforcement effect of GO and CIPs. As a result, the increment of the plateau stress resulting from the GO filling is minor. The plateau stress is even reduced with increasing the GO content from 0.1 to 0.5 wt.%, as shown in Figure 6b.

### 3.3. Dynamic Magnetorheological Tests

#### 3.3.1. Strain Amplitude Sweeps

Figure 9 shows the variations of G’ and tan(δ) versus the strain amplitude of the MREs under various magnetic fields. The value of tan(δ), which is the ratio of the loss modulus to the storage modulus (tan(δ) = G”/G’), is the measure of the damping property of the polymer composites. It can be observed that two regions could be divided from these plots. In Region I, G’ shows a very slow decline trend and tan(δ) almost keeps constant up to around 1% strain amplitude. This indicates that the microstructure is intact in this region. Then, in Region II, G’ starts to drop dramatically and tan(δ) rises rapidly with increasing the strain amplitude. These phenomena are associated with the Payne effect for filler-reinforced elastomers due to the breakdown and reformation of filler networks and the release of the rubber trapped in the filler networks [6,15,17,27,43,50]. The critical strain amplitude at which the severe drop happens in G’ is usually used to recognize the transition of MREs from a linear to nonlinear viscoelastic material [2,8,23].

It can be observed that, with the increase of magnetic field, the MREs show the increased storage moduli in Region I. This indicates that the obvious dynamic MR effect is present in all MRE samples. In Region II, the trend of the storage modulus decreasing with the strain amplitude is improved by increasing the magnetic field strength, as shown in Figure 9. This phenomenon is referred to as the magneto-induced Payne effect and is caused by the increased inter-particle distances with increasing strain, which in turn causes the rapid attenuation of magnetic dipolar interactions between the magnetic particles in MREs [1,2,15,16,51,52]. With a strain amplitude higher than 10%, a more severe magneto-induced Payne effect is observed due to the destruction and reformation of the magnetic particle network under the large deformation [6,51,52].

As for the loss factor, MRE-0 shows a different magnetic field dependence from those of the GO-filled MREs. As shown in Figure 9, the tan(δ) of MRE-0 increases with increasing the magnetic field in the whole the tested strain range. This may be due to the uniform distribution of CIPs and the high chemical crosslinking density of the PDMS matrix. Under the dynamic shear deformation, the particle networks deform and rotate synchronously with the applied shear deformation. Thus, the energy dissipation due to the interface friction between the particles and matrix is improved with increasing the magnetic field [3,27]. In Region II, more energy is dissipated by the destruction and reformation of the magnetic particle network due to the improved magnetic dipolar interactions [43,52]. Thus, the loss factor increases more rapidly with the strain amplitude under the higher magnetic field.

For the GO-filled MREs, the loss factors were almost magnetic field independent in Region I. The loss factors of MRE-1 and MRE-3 are 1.02 and 0.104, respectively, which is slightly less than that of MRE-0 (0.106) in the zero-field condition. The loss factor of MRE-5 is 0.176, much higher than the others. In Region II, the trend of loss factor increasing with the strain amplitude is obviously improved by increasing the GO content. With increasing the magnetic field, this trend is reduced for the strain amplitude less than about 20%. Then, it is greatly accelerated for the strain amplitude beyond 20%, as illustrated in Figure 9c. These results are caused by the reduced chemical crosslinking density and the increased aggregations of fillers (both CIPs and GO sheets) in the GO-filled MREs. In Region I, the deformation is mainly accommodated by the PDMS matrix between the fillers due to the reduced crosslinking density and the strong constraints of the fillers to the neighboring soft matrix. Thus, the energy dissipation primarily comes from the damping of the viscoelastic matrix. As a result, the loss factors of GO-filled MREs are independent of the magnetic field and less than that of MRE-0 in Region I. The high loss factor of MRE-5 is attributed to the severe filler aggregations, which induce great energy dissipation due to the filler–filler interactions under the dynamic shear loads. Then, in Region II, the energy dissipation increases with the GO content due to the breakdown of the physical crosslinking interfaces and the additional interfacial friction between the GO sheets and PDMS matrix. These factors cause the rapid increase of the loss factor. With increasing the magnetic field, the increased magnetic dipolar interactions between CIPs lead to the improved movements of CIPs, which induce more compact particle aggregations formed in the MREs and provide an increased constraint to the PDMS matrix. Thereby, the energy dissipation due to the damping of the viscoelastic matrix and the interface frictions is reduced by increasing the magnetic field strengths within the strain amplitude of 20%. For the strain amplitude beyond 20%, the energy dissipation corresponding to the destruction of the particle aggregations, debonding of the filler–matrix interfaces and reformation of magnetic particle network increase with the magnetic field [6,43,52], leading to the great increase in the loss factor with the strain amplitude.

To quantify the effect of GO content on the dynamic rheological and MR effect of the MREs, the strain amplitude at tan(δ) = 1, storage moduli at the strain amplitude of 0.01% and Payne effect of the MREs were extracted. Figure 10 shows the strain amplitudes of the MREs at tan(δ) = 1. It is known that, when the value of tan(δ) is greater than 1, the material exhibits a more fluid-like status. On the contrary, it exhibits a more solid status when the value of tan(δ) is less than 1 [23,49]. When 0 to 0.5 wt.% GO sheets are added, the strain amplitude at tan(δ) = 1 is significantly shifted from 90.83% to 31.37% in the zero-field condition, as illustrated in Figure 10. This means that the MRE sample filled with higher content of GO sheets exhibits more fluid-like behavior. This result is consistent with the analysis in Section 3.2 that the chemical crosslinking of PDMS chains is inhibited by the GO sheets. Thus, more linear structures rather than the chemical cross-linked structures of PDMS chains were produced in the MREs with higher GO content, which caused more fluid-like behavior.

With increasing the magnetic field, the strain amplitudes at tan(δ) = 1 of MRE-0 and MRE-1 show a decreasing trend, while those of MRE-3 and MRE-5 show an increasing trend before they reach magnetic saturation. This is because, in MRE-0 and MRE-1, the magnetic particles are uniformly dispersed in the relatively rigid matrix. Thus, small particle movements are induced in the MREs, and the contact pressure between the magnetic particles and PDMS matrix is improved with increasing the magnetic field. Then, more energy is dissipated due to the interface friction, resulting in the improved interface damping [3,17]. In MRE-3 and MRE-5, filler aggregations exist, as mentioned above. Thus, under the higher magnetic field, more compact magnetic particle aggregations are formed, which provides stronger constraint to the trapped matrix. Consequently, these constrained rubber and the compact particle aggregations improve the ability to store elastic energy, resulting in the decreased damping of MREs [2,5,11].

Figure 11 shows the storage moduli at the strain amplitude of 0.01% and Payne effect of the samples under various magnetic fields. The Payne effect was calculated by G’_γ=0.01%_ − G’_γ=60%_, where the storage modulus at the strain amplitude γ = 60% (G’_γ=60%_) is used to exclude the effect of the slipping between the sample and the loading plate of rotational rheometer under higher strain amplitudes. It can be seen that both the storage modulus and Payne effect increase with the GO content. The increased Payne effect is caused by the destruction of the increased filler aggregations and physical crosslinking interfaces between the GO sheets and PDMS matrix. The dynamic storage modulus also increases with the increase of GO content. In the zero-field state, G’ of MRE-0 is 0.29 MPa and the GO-induced increments are 0.045, 0.122 and 0.364 MPa for the samples filled with 0.1, 0.3 and 0.5 wt.% GO, respectively. In the magnetic saturation state, the G’ of MRE-0 is increased to 0.366 MPa due to the MR effect, while the GO-induced increments are 0.114, 0.191 and 0.536 MPa for the samples with 0.1, 0.3 and 0.5 wt.% GO, respectively. Comparing with those in the zero-field state, all GO-filled MREs show obvious improvement with the GO-induced increments in G’ in the magnetic saturation state. This is different from those found under the steady-state tests, where only MRE-1 shows the improved MR effect. The improved G’ in the GO-filled MREs may be caused by the dynamic self-stiffening effect occurring in the physical crosslinking interfaces around GO sheets. It has been reported that, under the dynamic loads, the dynamic self-stiffening effect could be resulted from the realignment and reorientation of PDMS chains along the surfaces of GO sheets, thus causing the improved physical reinforcement effect of GO [44]. Another possible mechanism is that the dynamic load promotes the movement of CIPs in the softer PDMS matrix, thus forming more regular particle chains and inducing the improved MR effect.

#### 3.3.2. Frequency Sweep

Figure 12 shows the storage moduli and loss factors of the MREs versus the test frequency in various magnetic fields. It is worth noting that the frequency sweep tests were carried out with constant strain amplitude of 0.1%, which is much less than the LVE strain limit as found in the strain sweep test. Therefore, all MRE samples exhibit more solid-like behavior and the values of tan(δ) are much less than 1. It can be observed that both the storage modulus and loss factor increase with the test frequency. The absolute MR effect of the storage modulus is not that sensitive to frequency, i.e., the curves of the storage modulus G’ versus the test frequency under various magnetic fields are nearly parallel to each other. These results are consistent with the findings of many experimental and theoretical studies and reflect the viscoelastic nature of elastomeric composites [2,3,11,15,23,27]. It is generally considered that the increase of the shear frequency leads to the enhancement in the chain entanglements, internal friction among the polymer chains and interfacial friction between the particles and rubber matrix, thus improving the storage modulus and loss factor [3,11,51]. The frequency independent MR effect is because the magnetic interaction forces between particles respond to the applied magnetic field promptly with hardly any time lag [15].

It also can be found that, in a given magnetic field, the storage modulus G’ shows the increasing trend with the increase of GO content due to the physical strengthening effect of GO sheets, which in turn leads to the decrement of loss factor since the loss modulus G” is not that sensitive to the GO content, as shown in Appendix A. For example, upon addition of 0.1–0.5 wt.% of GO sheets in MREs, the storage modulus of the MRE sample increases from 0.37 to 0.56 MPa (a growth of 51.3%) in the zero-field condition and from 0.49 to 0.75 MPa (a growth of 53.1%) at magnetic saturation. The loss modulus G” increases from 0.034 to 0.046 MPa (a growth of 35.3%) in the zero-field state and from 0.053 to 0.065 MPa (a growth of 22.6%) at magnetic saturation. The loss factor of MRE-0 shows an increasing trend with the increase of the magnetic field, while those of the GO-filled MREs are almost independent of the magnetic field when the frequency is higher than 1 Hz. These results are consistent with those of strain sweep tests.

#### 3.3.3. Dynamic MR Effect

To quantify the effect of GO content on the dynamic MR effect of the MREs, the absolute MR effect $\Delta G^{\prime }$ and the relative MR effect $\Delta {G^{\prime }}_{r}$ were calculated by Equations (1) and (2), respectively. The dynamic MR effects are plotted in Figure 13, in which the values of $\Delta G^{\prime }$ and $\Delta {G^{\prime }}_{r}$ were calculated based on the storage modulus at the strain amplitude of 0.1% and frequency of 1 Hz. It can be found that the absolute MR effect $\Delta G^{\prime }$ increases with the increase of GO content, while the highest relative MR effect is also achieved by MRE-1, which is improved by 10.8% from 28.7% to 39.5%. When the GO content is higher than 0.1 wt.%, the relative MR effect is reduced due to the increased zero-field properties and the increased aggregations of GO and CIPs. These results are consistent with those of steady-state tests. The improvement in the dynamic MR effect of the GO-filled MRE is due to the increased flexibility of the PDMS matrix and the dynamic self-stiffening effect occurring in the physical crosslinking interfaces around GO sheets.

## 4. Conclusions

To prepare the new MREs with the high MR effect and high zero-field stiffness, GO was used as the additive and isotropic PDMS-based MRE samples filled with different GO fractions of 0, 0.1, 0.3 and 0.5 wt.% were produced by a solution blending-casting method. The experimental studies were conducted to discover the effect of the GO sheets on the microstructure and the steady-state and dynamic MR effects of the GO-filled PDMS-based MREs.

The results reveal that, with the increase of GO content, a uniform distribution of GO sheets and CIPs becomes difficult and the aggregations of GO and CIPs increase. The strong physical crosslinking between the GO sheets and PDMS matrix induces the strong physical stiffness effect, which results in the increase of the zero-field modulus with the increase of GO content. The dynamic shear tests revealed that the MREs filled with higher GO content exhibit more fluid-like behavior. The addition of GO sheets inhibits the chemical crosslinking of PDMS chains, thus improving the flexibility of the PDMS matrix and endowing the magnetic particles with great mobility under the magnetic field. Under the steady-state shear load, MRE-1 shows the maximum absolute MR effect, and then it decreases with the GO content due to the increased aggregation of GO and CIPs. Under the dynamic shear load, the absolute MR effect increases with the GO content due to the increased flexibility of the PDMS matrix and the dynamic self-stiffening effect occurring in the physical crosslinking interfaces around GO sheets. The highest relative MR effect is achieved by the MREs filled with 0.1 wt.% GO sheets under both the steady-state and dynamic shear loads. Then, the relative MR effect decreases with the further increase of GO content due to the improved zero-field modulus and the increased agglomerations of GO and CIPs. The addition of GO sheets results in the reduction in the loss factor in the linear viscoelastic state, as well as a great increase in the nonlinear state due to the increased energy dissipation caused by the destruction of the particle aggregations, debonding of the filler–matrix interfaces and reformation of magnetic particle network under the magnetic field.

In summary, this study shows that the addition of GO sheets is a possible way to prepare new MREs with high MR effect, while simultaneously possessing high zero-field stiffness and load bearing capability. To further improve the MR effect of the GO-filled MREs, a more sophisticated fabrication process for dispersing the GO sheets and CIPs uniformly in MREs should be explored and the MR effect of the anisotropic GO-filled MREs will be studied in our future works.

## Figures and Tables

**Figure 1 polymers-13-01777-f001:**
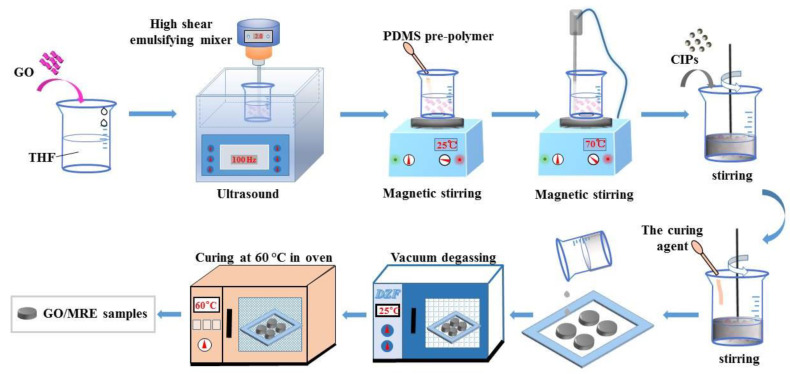
Schematic of the fabrication procedure for the isotropic GO-filled MREs.

**Figure 2 polymers-13-01777-f002:**
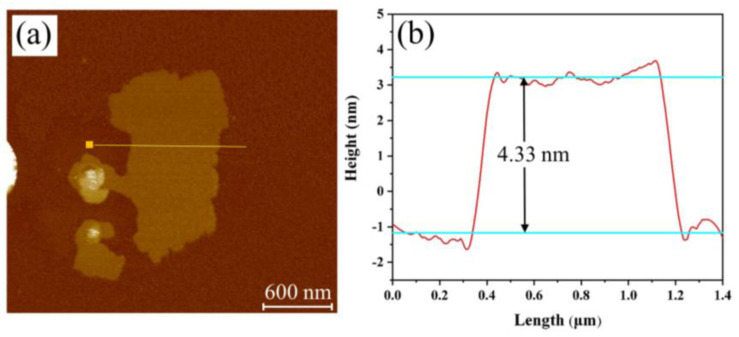
(**a**) Typical AFM image; and (**b**) the corresponding height profile of GO sheet.

**Figure 3 polymers-13-01777-f003:**
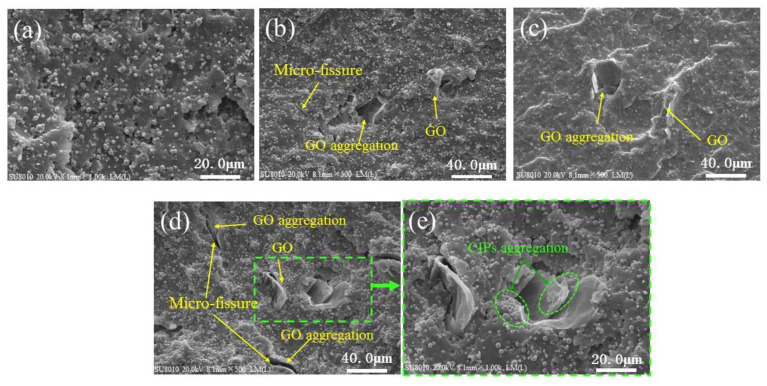
SEM images of: (**a**) MRE-0; (**b**) MRE-1; (**c**) MRE-3; (**d**) MRE-5; and (**e**) a locally magnified view of the fractured surface of MRE-5.

**Figure 4 polymers-13-01777-f004:**
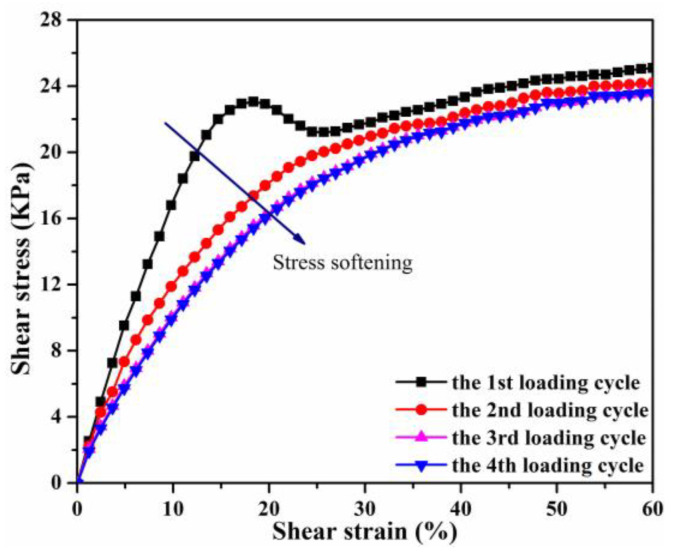
The steady-state shear stress–strain curves of MRE-0 under the repeated loading process in the zero-field condition.

**Figure 5 polymers-13-01777-f005:**
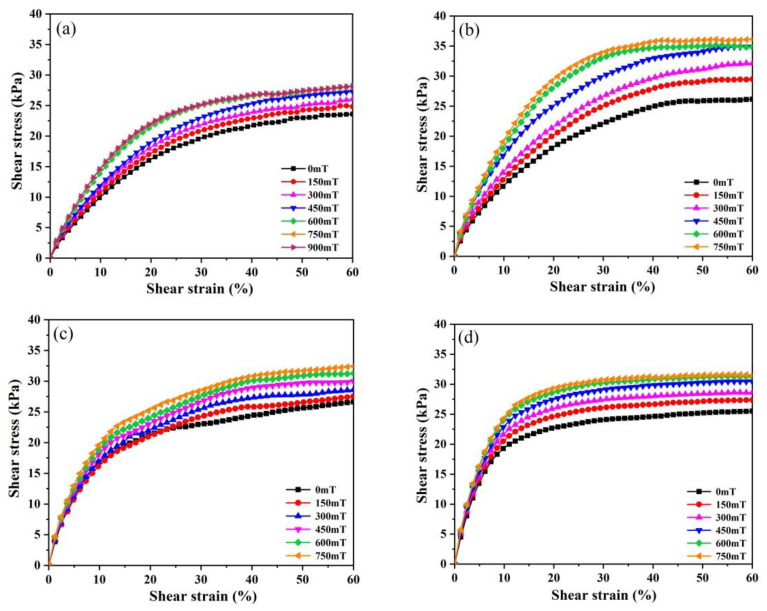
Steady-state stress–strain curves of the MREs under various magnetic fields: (**a**) MRE-0; (**b**) MRE-1; (**c**) MRE-3; and (**d**) MRE-5.

**Figure 6 polymers-13-01777-f006:**
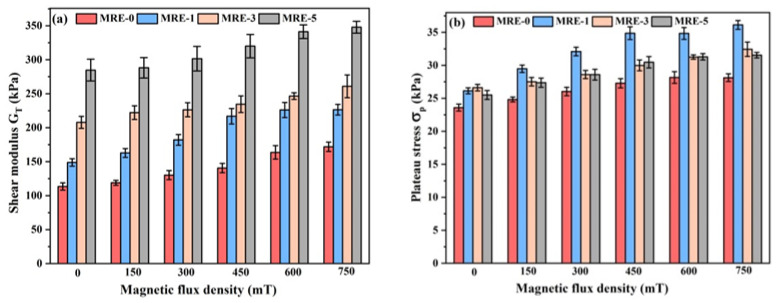
(**a**) Steady-state shear modulus; and (**b**) plateau stress of the MREs under various magnetic fields.

**Figure 7 polymers-13-01777-f007:**
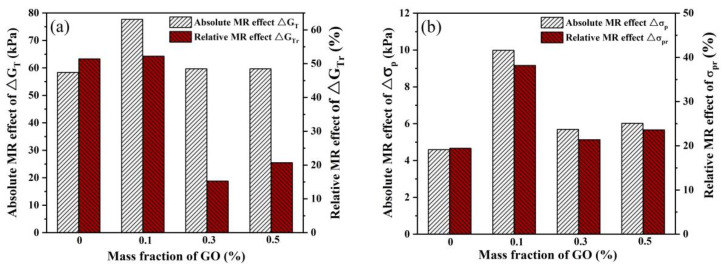
The absolute and relative MR effect of the MREs under steady-state shear load: (**a**) shear modulus; and (**b**) plateau stress.

**Figure 8 polymers-13-01777-f008:**
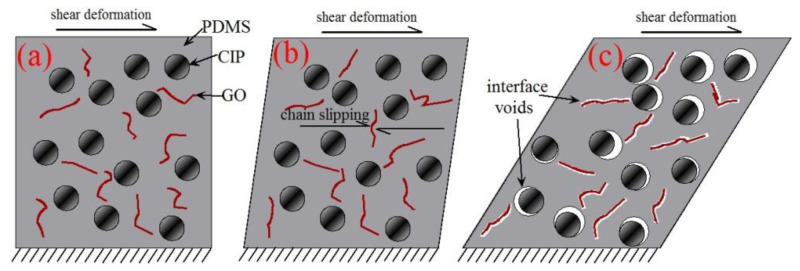
Schematic of the microstructural deformation mechanisms of GO-filled MREs under shear deformation: (**a**) initial state; (**b**) small shear strain; and (**c**) large shear deformation.

**Figure 9 polymers-13-01777-f009:**
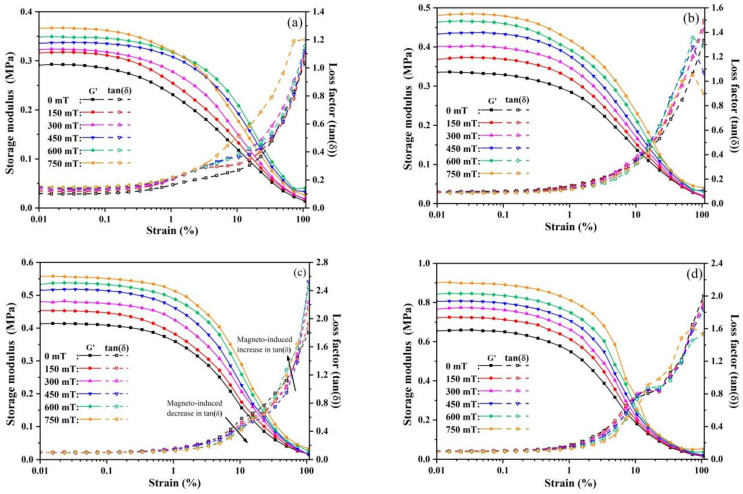
The storage modulus G’ and loss factor δ versus strain amplitude of the MREs under various magnetic fields: (**a**) MRE-0; (**b**) MRE-1; (**c**) MRE-3; and (**d**) MRE-5.

**Figure 10 polymers-13-01777-f010:**
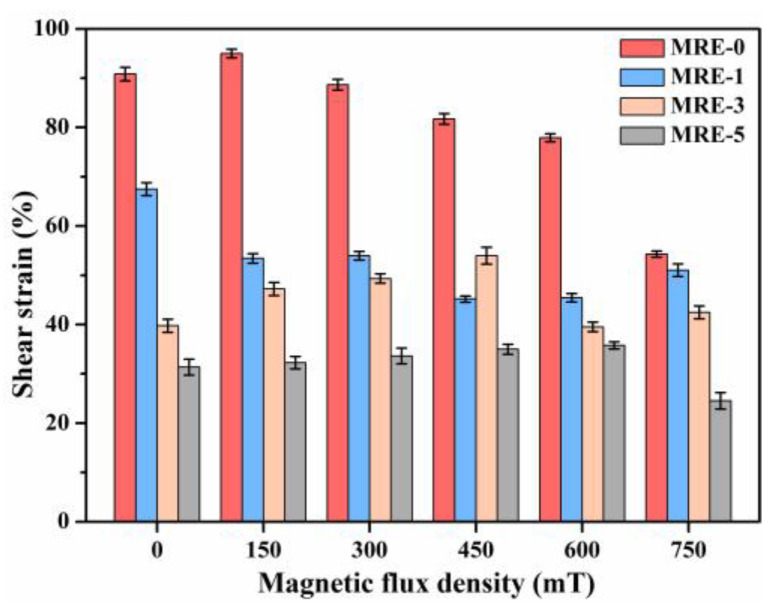
The strain amplitudes at tan(δ) = 1 of the MREs.

**Figure 11 polymers-13-01777-f011:**
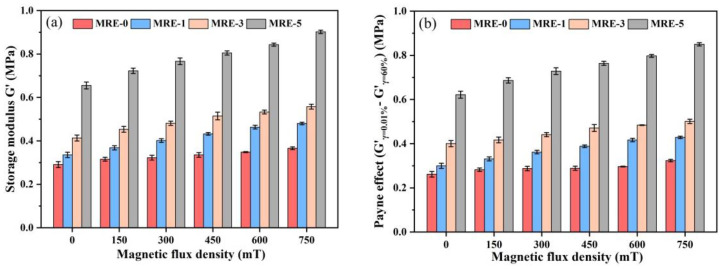
(**a**) Storage moduli at 0.01% strain amplitude; and (**b**) Payne effect (G’_γ=0_._01%_ − G’_γ=60%_) of the MREs under various magnetic fields.

**Figure 12 polymers-13-01777-f012:**
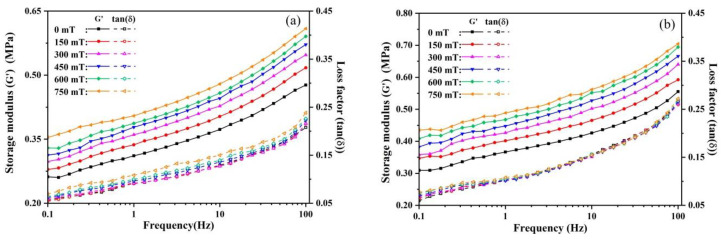
The plots of the storage modulus G’ and loss factor tan(δ) versus test frequency of the MREs under various magnetic fields: (**a**) MRE-0; (**b**) MRE-1; (**c**) MRE-3; and (**d**) MRE-5.

**Figure 13 polymers-13-01777-f013:**
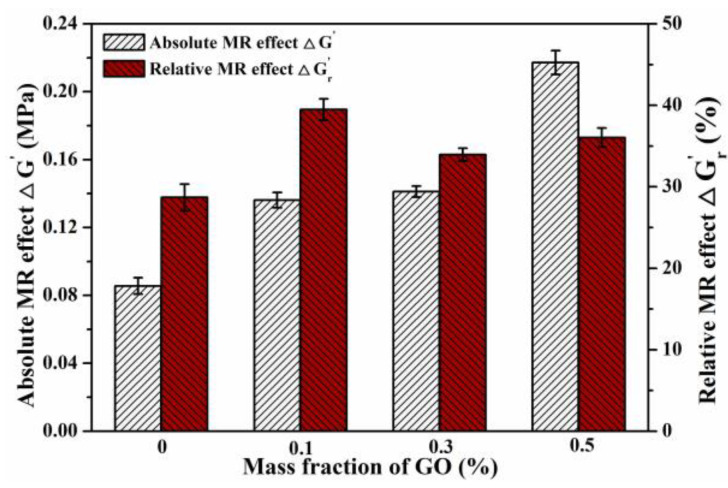
The absolute and relative MR effect of the MREs under dynamic shear loads.

## Data Availability

Not applicable.

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
