# Peer review of "Experimental Investigation on the Effect of Graphene Oxide Additive on the Steady-State and Dynamic Shear Properties of PDMS-Based Magnetorheological Elastomer"

_polymers, 2021, doi:10.3390/polym13111777_

Round 1
Reviewer 1 Report
- The authors should give more precise details about the purpose of the work in the introduction section. The novelty or significance of this work is not clearly stated.
- Literature needs to be updated in the manuscript.
- Page 6- Figure 5e? must be corrected?
- Many space errors/punctuation errors must be solved. The abbreviations should be checked in the manuscript and make clear. Several times the parameters are defined in the manuscript.
- Figure 5,6, 8, 9,12- why need to define (a) several times?
- There are also some grammar issues in the text. The authors are required to make a revision throughout the whole manuscript to improve the English writing thoroughly and carefully.
- Expand the conclusions in relation to the specific goals and the future work.
Reviewer 2 Report
Thank you for the opportunity in reviewing the manuscript (polymers-1223518). Interesting results are reported and the manuscript was reviewed for publication in Polymers-MDPI Journal. However, the paper require substantial revision before acceptance and consideration for publication. Few points are -
- In paper, what is the reason for choosing graphene oxide nanosheets as secondary reinforcing filler to reinforce properties in CIP based MREs? Infact, CNT would have been a better choice due to its high aspect ratio and high reinforcing ability than graphene oxide?
- The terminology of using “graphene oxide nanosheets” in paper the thickness of 10-20 nm should be cross checked. Generally, graphene oxide is a monolayer material with a high oxygen content, typically characterized by C/O atomic ratios less than 3.0 and typically closer to 2.0 [1].
[1] https://doi.org/10.1016/j.carbon.2013.08.038.
- Introduction is well-described. However, to cover diversity of work, authors need to cite following related articles: (1) 10.1109/TMAG.2013.2275736; (b) https://doi.org/10.1002/pat.4544 (c) https://doi.org/10.1016/j.jmmm.2019.03.075.
- In #section 2.2 (fabrication of MREs samples), how authors make sure the process of complete evaporation of THF from composite? Normally, the presence of THF in MREs specimen may act as diluent and affects the properties? Please clarify?
- In Figure 2, how many replicas of the AFM were performed? Is this mean or average value of thickness?
- In Figure 3, the SEM images are fizzy, the magnification scale is not visible. However, hypothesis of description of SEM images seems not that scientific. Please cross check. Reviewer thinks that SEM images should be shown at high resolution scale.
- Figure 5 and 6 seems shows anisotropic effect in MREs based on GO/CIP with increasing GO content? BUT reviewer wonder is it the increase in shear stress and shear modulus is due to effect of orientation of CIP particles in increasing magnetic field or due to increasing reinforcement of GO or both?
- Measurements on isotropic specimens seems not performed by the authors. Then, how they determine magnetic effect in Figure 8?
- Similarly to Figure 5,6, In Figure 11, is the higher values of MRE-5 specimen at all magnetic fields due to higher orientation of CIP particles or reinforcing effect of GO is not clear? How MR effect determined in Figure 13 without performing isotropic studies?
Round 2
Reviewer 2 Report
Major comment -
- The scale bar of Figure 3 is NOT visible. Please make it bold and visible.
- To clarify the increase in modulus either due to GO reinforcement or orientation of CIP. Authors need to perform control experiments of PDMS composites based on GO and PDMS composites based on CIP under magnetic field? Then, situation may be clear which factor contribute the modulus and how much?
